

**Investigating the Mechanism of Typhoon Tracks on Ozone Pollution**
**Episodes in Guangdong, China**
**Xi Chen[1],Xiaoyang Chen[2*],Long Wang[1],Shucheng Chang[1],Minhui Li[1],Chong**
**Shen[3],Chenghao Liao[1],Yongbo Zhang[1],Mei Li[4], Xuemei Wang[5] ***
1. Institute of Atmospheric Environment, Guangdong Provincial Academy of
Environmental Science, Guangzhou 510045, China
2. Guangzhou Institute of Tropical and Marine Meteorology of China Meteorological
Administration, GBA Academy of Meteorological Research, Guangzhou, 510640,
China
3. Guangzhou Ecological and Agricultural Meteorological Center, Guangzhou, 511430,
China
4. Institute of Mass Spectrometry and Atmospheric Environment, Guangdong
Provincial Engineering Research Center for On-line Source Apportionment System
of Air Pollution, Jinan University, Guangzhou 511486, PR China
5. Guangdong-Hongkong-Macau Joint Laboratory of Collaborative Innovation for
Environmental Quality, College of Environment and Climate Jinan University,
Guangzhou 511486, PR China
Corresponding author: Xuemei Wang (eciwxm@jnu.edu.cn), Xiaoyang Chen
(chenxiaoyang@gd121.cn)





**Key Points:**
• Proximal northward-recurving typhoons are the most likely to induce ozone
pollution.
• The northward typhoon will cause ozone to increase by 0.3~12.3ppbv in
vertical height.
• The contribution rate of transboundary layer transport under the influence of
typhoon to the ozone in the boundary layer can reach 16%.



**Abstract**

Ozone ($O_3$) pollution has emerged as one of the core challenges in atmospheric
environmental governance in China, particularly in Guangdong Province. As a crucial
weather system during East Asian summers, typhoons exert profound influences on $O_3$
formation, accumulation, and transboundary transport through variations in their
tracks and intensities. This study examined 237 historical typhoons occurring in China's
coastal waters between 2013-2023, classifying them into three distinct trajectory
types using k-means clustering: westward-moving typhoons (Type 1), Distant
northward-recurving typhoons (Type2) and Proximal northward-recurving typhoons
(Type3). By integrating ground-based observations, reanalysis data, and WRF-CMAQ
model simulations to investigate the mechanisms through which typhoon tracks affect
ozone pollution in Guangdong Province. The results demonstrate that for Guangdong
Province, proximal northward-recurving typhoons induce more extreme
meteorological conditions compared to westward-moving and distant northward-
moving typhoons. Backward trajectory analysis reveals that northward-moving
typhoons significantly enhance vertical downward transport of upper-level ozone,
increasing ozone vertical gradients in Guangdong Province, with concentration
enhancements of 2.5–11.6 ppbv (Type 2) and 0.3–12.3 ppbv (Type 3). The analysis of
consecutive northward-moving typhoons' impact on ozone pollution in Guangdong
Province reveals that surface photochemical reactions served as the dominant factor,
while vertical downward transport of upper-level ozone acted as a secondary
contributor. During this event, vertical transport contributed up to 39.9 ppbv to near-
surface (100 m) ozone concentrations, with cross-boundary-layer transport accounting
for up to 16% of boundary layer ozone concentrations, demonstrating that typhoon-
induced vertical transport significantly enhances boundary layer ozone levels and
consequently worsens surface pollution.



**Plain Language Summary**

It is well established that typhoon tracks exert significant impacts on ozone pollution. However, current research predominantly focuses on individual typhoon case studies or isolated meteorological factors, leaving a gap in comparative analyses of the mechanisms associated with different typhoon pathways. This study categorizes the trajectories of 237 typhoons that occurred over the western Pacific Ocean, specifically investigating the influence mechanisms of westward-moving typhoons (Type1), distant northward-moving typhoons (Type2), and Proximal northward-recurving typhoons (Type3) on ozone pollution in Guangdong Province. The results demonstrate that close-in northward-moving typhoons induce the most favorable conditions for ozone formation and the least favorable atmospheric dispersion conditions in Guangdong, thereby promoting ozone pollution. Additionally, northward-moving typhoons facilitate the subsidence of high-latitude, high-concentration ozone into the boundary layer, leading to elevated ozone levels. Finally, consecutive northward-moving typhoons trigger widespread and persistent ozone pollution across Guangdong. During this process, cross-boundary-layer transport via vertical motion contributes up to 16% of the ozone concentration within the boundary layer, underscoring the substantial impact of northward-moving typhoons on boundary-layer ozone through vertical transport mechanisms.

**1 Introduction**

Ozone ($O_3$) pollution has become one of the core challenges in atmospheric environmental governance in China, particularly in the Pearl River Delta region. As a typical secondary pollutant, the formation of $O_3$ is dually regulated by precursor emissions (NOx and VOCs) and meteorological conditions. (Dou et al., 2024; Gong et al., 2025; Qiu et al., 2025; Yang et al., 2019)。 In recent years, despite continuous strengthening of anthropogenic emission control measures, the increasing frequency of extreme weather events has significantly amplified the complexity of $O_3$ pollution.



(Chen et al., 2022a; Lu et al., 2024; Wan et al., 2022; Wang et al., 2024a)。 Among
these factors, typhoons, as a crucial weather system during the East Asian summer,
exert profound impacts on ozone (O₃) formation, accumulation, and transboundary
transport through their track and intensity variations, which significantly modify
regional meteorological conditions (e.g., temperature, humidity, wind speed) and
atmospheric transport processes. (Chen et al., 2021; Qu et al., 2021; Shen et al., 2023;
Wang et al., 2022a)。
The peripheral subsidence flows of typhoons frequently induce high
temperatures, low humidity, and stagnant weather conditions, which enhances
photochemical reactions while suppressing pollutant dispersion, consequently leads
to localized O₃ accumulation. (Chen et al., 2022b). Simultaneously, the heat stagnation
induced by typhoon conditions favors biogenic emissions, with isoprene
concentrations potentially doubling, leading to BVOCs contributing up to 10 ppbv to
O₃ formation (Xu et al., 2023). Under the combined influence of the western Pacific
subtropical high and typhoon peripheral circulation, tropical cyclones facilitate the
downward transport of ozone-rich stratospheric air into the lower troposphere. This
process leads to the formation of elevated ozone concentrations in the middle
boundary layer. Subsequently, the downward transport of residual layer pollutants
significantly contributes to the accumulation of ground-level pollutant concentrations
during morning hours (Chen et al., 2021; Zhan et al., 2020; Chen et al., 2022c). The
approach of typhoons significantly enhances both biogenic emissions and
transboundary ozone transport, with observed increases reaching 78.0% and 22.5%,
respectively. The abundance of precursors coupled with intensified photochemical
reactions more than doubles ozone formation efficiency.(Wang et al., 2022a).
Regarding the effect of vertical transport on ozone pollution under typhoon weather,
some researchers attribute observed ozone increases primarily to enhanced surface-
level photochemical activity (Huang et al., 2021; Jiang et al., 2024; Wang et al., 2025).
In this view, upper-level subsidence flows primarily suppress tropospheric ozone



dispersion without making significant positive contributions to ozone transport (Ding
et al., 2023; Li et al., 2022; Ouyang et al., 2022). Other studies have documented cases
where typhoon-induced vertical mixing facilitates downward transport of elevated
ozone layers (e.g., through stratospheric intrusions), generating measurable surface
ozone enhancements of 10-15 ppbv (Chen et al., 2021).
Current research demonstrates that typhoon impacts on ozone pollution exhibit
significant path dependence. Westward-moving typhoons induce increased net ozone
production in the Pearl River Delta (PRD) core region prior to landfall, followed by a
rapid decline to near-zero levels on the landfall day(Ding et al., 2023). During their
initial development stage, typhoons enhance ozone production through subsidence-
induced meteorological conditions. However, as they approach landfall, associated
heavy precipitation and strong winds effectively scavenge pollutants, leading to
negative ozone anomalies over the Yangtze River Delta region. These anomalies extend
vertically, with maximum ozone reductions of 14-18 ppbv observed at 5 km
altitude(Chen et al., 2021). When typhoons track northward across the Taiwan Strait
through the low-latitude western Pacific, they trigger sequential regional ozone
pollution episodes in both the Yangtze River Delta (YRD) and Pearl River Delta (PRD)
regions(Wang et al., 2022b). The northerly peripheral circulation of such typhoons
transports precursors from North China and the Yangtze River Delta (YRD) southward,
which, when superimposed with local emissions, triggers abrupt ozone concentration
increases(Shen et al., 2023). Successive northward-moving typhoons can elevate $O_3$
concentrations by 30% across eastern China while prolonging pollution duration(Wang
et al., 2024b). Furthermore, the interaction between typhoons and the subtropical
high can form a compound weather system, which exacerbates $O_3$ pollution intensity
and prolongs its duration(Gao et al., 2020; Han et al., 2020a; Qin et al., 2020). However,
current research predominantly focuses on individual typhoon cases or isolated
meteorological factors(Kumar et al., 2023; Li et al., 2023a; Zhan et al., 2020), leaving
significant gaps in comparative analyses of mechanisms associated with different



typhoon tracks. Key unresolved questions include: How do typhoons with distinct
paths differentially modulate meteorological conditions and regional transport? How
do large-scale circulation changes induced by varying typhoon tracks influence the
vertical distribution of ozone? These questions demand systematic investigation to
advance our understanding of typhoon-$O_3$ interactions.
As a high-frequency typhoon landing region, Guangdong Province exhibits
particularly strong correlations between ozone pollution and typhoon activity
(Shuping et al., 2022; Yaoyao et al., 2022). Statistical analyses reveal that over 80% of
ozone exceedance days during Guangdong's summer-autumn seasons from 2015-
2021 were typhoon-associated (Shen et al., 2023). Under climate change scenarios,
observed trends of northward-shifting typhoon tracks and intensifying storm strength
may further alter regional ozone pollution patterns (Guo and Tan, 2022). Consequently,
elucidating the mechanistic links between typhoon paths and ozone pollution holds
dual significance: advancing regional atmospheric multipollutant theory while
providing scientific foundations for dynamic, precision-based ozone control strategies.
This study systematically investigates all typhoons near Guangdong Province from
2013 to 2023 by integrating multi-source observational data and numerical
simulations. Through comprehensive classification of typhoon tracks, we conduct in-
depth analyses of the relationships between meteorological factors, circulation
patterns, atmospheric transport, and three-dimensional ozone distribution under
different typhoon paths. Specifically, we examine the contribution of upper-level
transport to boundary layer ozone concentrations during typical typhoon events. The
research aims to elucidate the differential impacts of various typhoon tracks on $O_3$
pollution in Guangdong region, thereby providing scientific support for refined air
quality management strategies.
**2 Materials and Methods**
2.1 K-means Clustering Analysis
K-means represents one of the most prevalent partition-based clustering



methods. The algorithm categorizes n objects into K clusters based on a predefined
parameter K, aiming to minimize the within-cluster sum of squares (WCSS) while
maximizing the between-cluster sum of squares (BCSS). This ensures high intra-cluster
similarity and low inter-cluster similarity. The K-means algorithm has been widely
applied in atmospheric trajectory classification studies due to its effectiveness in
identifying characteristic transport patterns(Han et al., 2020b; Yufeng et al., 2024; Zhu
et al., 2023).

In this study, we performed two distinct clustering analyses using the K-means

method: typhoon track clustering and atmospheric transport pathway clustering. For
typhoon track clustering: 1. Targeted typhoon tracks over the western Pacific Ocean;
2.Employed Euclidean distance metric for data point allocation; 3. Determined the
optimal K value by identifying the elbow point where the rate of WCSS decrease
substantially diminished; 4. Selected K=3 as the optimal cluster number, yielding three
distinct typhoon track types (**Fig.S3**). For atmospheric transport pathway clustering:1.
Analyzed 7-day three-dimensional backward trajectories; 2.Classified atmospheric
transport channels into four categories  (**Fig.S4**); 3.Implemented similar optimization
procedures for cluster determination. The methodology ensures statistically robust
classification of both typhoon trajectories and associated air mass transport patterns,
providing a quantitative basis for subsequent ozone transport analysis.

2.2 HYSPLIT Trajectory Model

HYSPLIT is a complete system for computing simple air parcel trajectories, as wel

l as complex transport, dispersion, chemical transformation, and deposition simulatio
ns. A common application is a back trajectory analysis to determine the origin of air m
asses and establish source-receptor relationships(Rolph et al., 2017; Stein et al., 201

5).

This study employs the NOAA HYSPLIT Trajectory Model (https://www.ready.noa

a.gov/HYSPLIT_traj.php) to conduct backward trajectory simulations for 237 typhoon





s in the Western Pacific region between 2013 and 2023. The meteorological data used
is GDAS (1-degree resolution). The source location is set at 113.5°E, 23.6°N, with the
backward trajectories initiated at 14:00 (local time) on the day of peak ozone pollutio
n during each typhoon event. The backward simulation runs for 168 hours (7 days), w
ith trajectory heights set at 500 m, 1000 m, and 2000 m above ground level (AGL).

2.3 WRF-CMAQ
The WRF-CMAQ modeling system was employed to simulate meteorological
fields and ozone concentration variations during the typhoon process. The WRF
(Weather Research and Forecasting) model version 3.9 was configured with the
following parameterizations: Microphysics,WSM6 Scheme;Cumulus
Parameterization:Grell-Freitas (GF) Scheme; Radiation:RRTMG Scheme;
Boundary Layer:YSU Scheme;Surface Layer:MM5 Similarity Theory;Land
Surface:Noah LSM. The large-scale meteorological fields and boundary conditions
were derived from NCEP's Global 6-hourly FNL forecast data. The CMAQ (Community
Multiscale Air Quality) model version 5.0.2 was implemented with the IPR (integrated
process rate) analysis module. The CB05 mechanism was selected for gas-phase
chemistry, while the AE6 mechanism was adopted for aerosol chemistry.
The modeling system utilized a triple-nested grid configuration (see **Fig.S1**) with
Lambert conformal projection centered at 114°E, 28.5°N and two standard parallels at
15°N and 40°N. The outermost domain (D01) had a horizontal resolution of 27 km ×
27 km, covering China, Southeast Asia and the western Pacific region. The
intermediate domain (D02) featured a 9 km × 9 km resolution encompassing South
China, while the innermost domain (D03) employed a 3 km × 3 km resolution focusing
on Guangdong Province and surrounding cities. The vertical structure consisted of 14
layers with the model top set at 200 hPa. For the first and second nested domains, the
air pollutant emission inventory adopted was the 0.25°×0.25° MEIC (Multi-resolution
Emission Inventory for China) developed by Tsinghua University for the year 2020. For



the third (innermost) domain, a higher-resolution 3 km×3 km emission inventory
compiled by the research team (Li et al., 2023b) was utilized. The simulation period
spanned from 00:00 UTC on 24 August to 00:00 UTC on 31 August 2020.
In the present study, $O_3$ was used as a model pollutant to analyze the effects of
atmospheric processes on the pollutants' value in deep convection events by using
Integrated Process Rate (IPR) analysis. The IPR analysis in CMAQ can be used to
calculate the influence of different atmospheric processes on the values of pollutants,
and to quantify the importance of each process in the evolution of the pollutant
value(Chen et al., 2018; Chen et al., 2022a). The processes include gas-phase chemistry
(CHEM),vertical advection (ZADV), horizontal advection (HADV), vertical diffusion
(VDIF), horizontal diffusion (HDIF), dry deposition(DDEP) and cloud processes (CLDS).
**3 Data**
3.1 Typhoon track data
The typhoon track data were obtained from the CMA Best Track Dataset
(tcdata.typhoon.org.cn) maintained by the Tropical Cyclone Data Center of China
Meteorological Administration. This dataset provides 6-hourly positional and intensity
records of tropical cyclones in the Northwest Pacific (including the South China Sea,
north of the equator and west of 180°E) since 1949, covering all typhoons
approaching/making landfall in China, with a spatial resolution of 0.1°×0.1° (Lu et al.,
2021; Ying et al., 2014). For this study, we extracted all typhoon track data from
January 1, 2013, to December 31, 2023, including temporal, geographical coordinates
(longitude and latitude), and intensity information. After interpolating the data, we
performed typhoon track classification using the K-means clustering method.

3.2 Ozone data
The ground-level ozone monitoring data were obtained from the China National
Environmental Monitoring Center (CNEMC). This dataset contains hourly
concentrations of $SO_2$, $NO_2$, CO, $O_3$, $PM_{10}$, and $PM_{2.5}$ from 1,657 monitoring stations



across China. For this study, we extracted hourly $O_3$ data from 105 stations within
Guangdong Province (station locations are shown in **Fig.S2**). Following the "Technical
Regulation on Ambient Air Quality Index (on trial)" (HJ 663-2013), we calculated the
daily maximum 8-hour average ozone concentration (MDA8 $O_3$). Days with MDA8 $O_3$
concentrations exceeding 160 μg/m³ (approximately 75 ppbv) were identified as ozone
exceedance days.

The TROPESS Chemical Reanalysis $O_3$ Increment 6-Hourly 3-dimensional Product

V1 dataset from NASA was utilized to investigate the three-dimensional spatial
distribution       of      ozone       under       typhoon       conditions
(https://disc.gsfc.nasa.gov/datasets/TRPSCRO3I6H3D_1/summary). The data are part
of the Tropospheric Chemical Reanalysis v2 (TCR-2) for the period 2005-2021. TCR-2
uses JPL's Multi-mOdel Multi-cOnstituent Chemical (MOMO-Chem) data assimilation
framework that simultaneously optimizes both concentrations and emissions of
multiple species from multiple satellite sensors. The data files contains a year of data
at 6-hourly resolution, and a spatial resolution of 1.125 x 1.125 degrees at 27 pressure
levels between 1000 and 60 hPa. This study extracted data from January 1, 2013 to
December 31, 2021 for spatial analysis of ozone distribution.

3.4 Meteorological data

Meteorological data from ERA5 (the fifth-generation European Mesoscale

Weather Forecasting Center reanalysis of global climate and weather for the past four
to seven decades) was also adopted in order to understand the pollution
characteristics. The temporal resolution of the data is hourly and the spatial resolution
is 0.25° × 0.25°. The parameters extracted herein include 2-m temperature, surface
relative humidity, total cloud cover, downward UV radiation at the surface, total
precipitation, mean sea level pressure, the u-component and v-component of wind at
the 10m, 175hPa and 900hPa level, boundary layer height, vertical velocity at the 850
hPa     level,     the     Geopotential     at     the     175hPa     and     900hPa     level.





(https://cds.climate.copernicus.eu/cdsapp#!/dataset/reanalysis-era5-single-
levels?tab=overview).

3.5 ground-level ozone reanalysis dataset

The ground-level MDA8 $O_3$ concentrations across China were obtained from the

China 1km High-Resolution Daily Ground-Level Ozone ($O_3$) Dataset (2000–2023), a
high-resolution (1 km) product developed by Wei et al. and hosted on the National
Earth System Science Data Sharing Platform (http://geodata.nnu.edu.cn) (Wei et al.,
2022). This dataset was generated through an ensemble learning approach combining
multi-source data, including hourly $O_3$ measurements from ~940 to 1,630 monitoring
stations (2013–2020) under China's Ministry of Ecology and Environment (MEE)
network, OMI/Aura total-column $O_3$ and tropospheric $NO_2$ retrievals, downward solar
radiation (DSR) and surface air temperature (TEM) from ERA5 reanalysis (0.1°
resolution), emissions of $NO_x$, VOCs, and CO from MEIC inventory, land cover from
MODIS, elevation from SRTM, and population density from LandScan. The subset of
data from January 1, 2013, to December 31, 2023, was temporally aligned with
recorded typhoon tracks to assess the spatio-temporal variability of $O_3$ during periods
with distinct typhoon track types.
**4 Results**
4.1 Characteristics of ozone pollution under different typhoon paths
4.1.1 Typhoon track clustering

Through k-means clustering analysis, the 237 typhoon tracks over the western

Pacific from 2013 to 2023 were classified into three distinct types (**Fig1.a-c**). Type 1
comprises typhoons that form in the western Pacific, move into the South China Sea,
and subsequently make landfall in South China or pass through its southern maritime
areas (total: 105 cases). Type 2 consists of typhoons originating from low-latitude
regions of the western Pacific that approach China before recurring northward,





traversing Japan and Korea before returning to the western Pacific basin (total: 77
cases). Type 3 represents typhoons generated in low-latitude western Pacific regions
that approach China and recurve northward, ultimately making landfall in China or
dissipating near Japan/Korea (total: 55 cases).

For clarity, these three typhoon types are respectively designated as: Type 1:

Westward-moving typhoons; Type 2: Distant northward-recurving typhoons; Type 3:
Proximal northward-recurving typhoons. Temporal distribution analysis (**Fig1.d**)
reveals that both Type 1 and Type 2 primarily occur from July to November, with peak
frequency in autumn, while Type 3 is predominantly observed from July to September,
showing maximum occurrence during summer.
4.1.2 Characteristics of ozone pollution

Figures 1a-1c illustrate the temporal evolution of maximum daily 8-hour average

ozone (MDA8) concentrations in Guangdong Province in relation to typhoon track
movements. From the perspective of ozone pollution characteristics, during the
approach of Type 1 typhoons toward mainland China, ozone concentrations in
Guangdong Province exhibit a gradual increase. If the typhoon does not make landfall,
ozone concentrations remain elevated until the typhoon dissipates. However, if the
typhoon makes landfall, ozone concentrations decrease rapidly due to precipitation
and strong winds (**Fig. 1a**). Recent studies highlight the dual effects of typhoons on
ozone: initial stages often enhance ozone through photochemical processes and
stratospheric intrusions, whereas landfall phases suppress it via convective activity and
precipitation(Chen et al., 2021; Li et al., 2021). Typhoons of Type2 can induce ozone
concentration increases in Guangdong Province through long-distance influences, as
demonstrated by cases where northward-moving typhoons beyond 40°N still triggered
ozone elevation in Guangdong (**Fig.1b**). This phenomenon may be associated with
large-scale transport of ozone and its precursors. Typhoons of Type3 tend to induce
ozone pollution in Guangdong when approaching eastern China, with peak
ozone concentrations occurring when the typhoon reaches approximately 25°N



latitude. Following typhoon landfall or eastward deflection, ozone
concentrations decrease (**Fig. 1c**).
We extracted the MDA8 O$_3$ concentrations during each typhoon event and
calculated Type-specific averages to examine ozone distribution patterns in
Guangdong under different typhoon types (**Fig. 2**). The results demonstrate that: Type
1 corresponds to MDA8 O$_3$ concentrations ranging 9.2-70.9 ppbv, with an average of
20 monitoring stations exceeding standards. Type 2 shows MDA8 O$_3$ concentrations of
12.2-90.3 ppbv, averaging 34 exceedance stations. Type 3 exhibits MDA8 O$_3$
concentrations of 3.3-89.7 ppbv, with 35 stations exceeding limits on average. The
spatial analysis reveals that ozone hotspots for all types consistently cluster in central
Guangdong, indicating similar spatial distribution patterns despite varying intensity.
Type3 exhibited the highest number of non-compliant monitoring sites, while Type1
showed the lowest count.

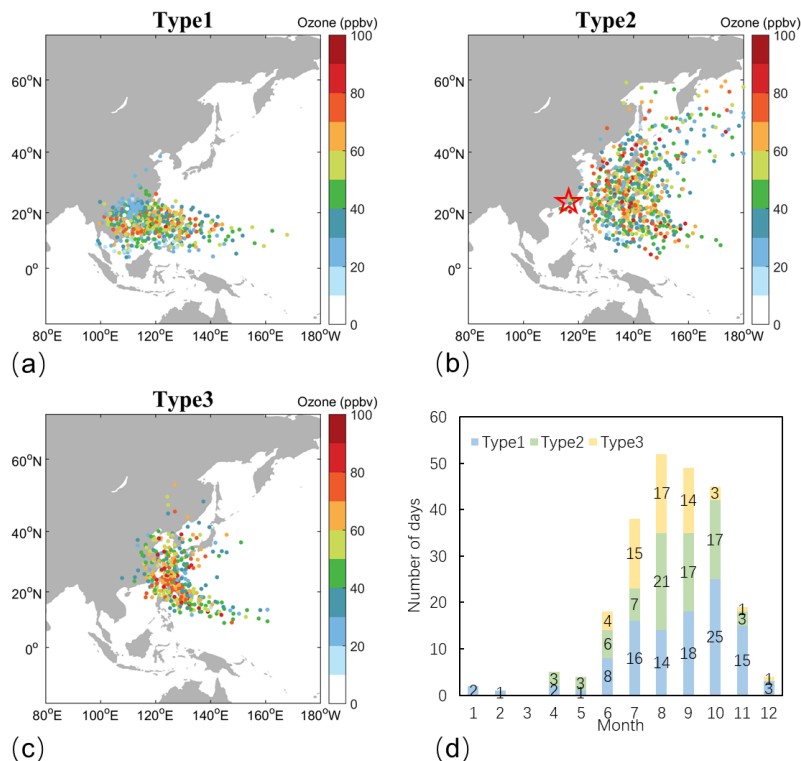





Figure 1. (a-c) Maximum daily 8-hour average (MDA8) ozone concentrations in Guangdong
Province (marked by red pentagrams) under different typhoon tracks(Different colors of dots
represent the average ozone concentration at all monitoring stations in Guangdong Province
when the typhoon is at that location), and (d) the corresponding temporal distributions of
typhoon occurrences for each track type.

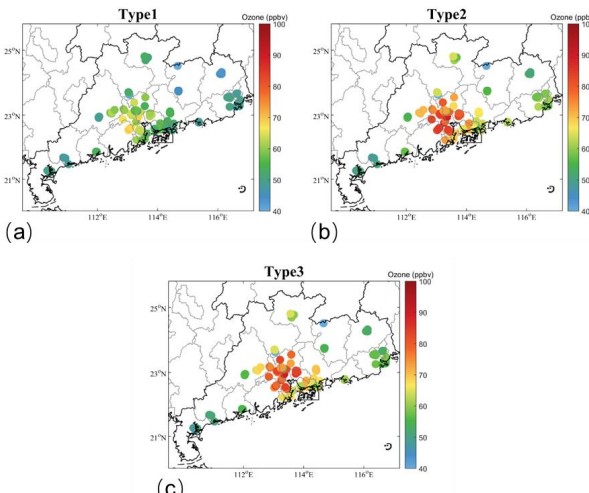


Figure. 2 Distribution of ozone pollution under different typhoon paths
4.1.3 Meteorological characteristics
To investigate the influence of meteorological factors on ozone pollution in
Guangdong Province under different typhoon tracks, we compared the differences in
meteorological conditions between three types of typhoon weather and non-typhoon
weather in Guangdong. Typhoon conditions refer to the day with the most severe
pollution during the typhoon event, while non-typhoon conditions correspond to the
remaining periods after excluding the entire typhoon process. The meteorological
factors analyzed included surface temperature, total cloud cover, surface solar
radiation, precipitation, surface relative humidity, boundary layer height, 10m wind
speed, and vertical velocity at 850 hPa.  All meteorological data were extracted from
ERA5 at 14:00 local time for comparative analysis.
The results indicate that, compared to non-typhoon weather, typhoon weather
in Guangdong is characterized by higher temperatures, stronger solar radiation, lower
cloud cover, reduced precipitation, lower relative humidity, higher boundary layer



height, weaker surface winds, and suppressed vertical motion (**Fig.3**). The peripheral
circulation of typhoons modifies the thermodynamic and dynamic structure of the
boundary layer, creating an "ideal reactor" for ozone formation. Near-surface
conditions of high temperatures, low humidity, and weak winds foster a stable
boundary layer structure, significantly enhancing photochemical reaction rates (Ding
et al., 2023). Additionally, increased solar radiation and elevated boundary layer height
further expand the spatial domain for ozone production.
A comparison of meteorological characteristics across different typhoon track
types reveals that Type 3 corresponds to what may be termed "extreme"
meteorological conditions. It brings high temperature(32.4℃), high radiation
(0.28MJ/m²), low cloud cover (0.47), low precipitation (0.08mm), low relative humidity
(65.4%), high boundary layer height (1.03km), low wind speed(2.08m/s), and less
vertical movement (-0.02pa/s) meteorological conditions, which are more likely to
cause ozone pollution in Guangdong Province. Compared to non-typhoon conditions,
Type 3 exhibits a temperature increase of 7.6°C, a cloud cover reduction of 0.28, a
radiation intensity enhancement of 0.09 MJ/m², and a boundary layer height elevation
of 0.21 km. It demonstrates the poorest horizontal diffusion conditions, with a near-
surface wind speed of 1.12 m/s lower than non-typhoon conditions. The severe ozone
pollution observed in Guangdong Province results from the combined effects of strong
ozone production rates and poor diffusion conditions, creating a synergistic
amplification of pollution levels. The photochemical reaction conditions in Type2 are
slightly weaker than those in Type3; however, reduced precipitation inhibits the wet
scavenging of ozone and its precursors. Additionally, strong subsidence at the 850 hPa
level not only suppresses the vertical diffusion of pollutants within the boundary layer
but also transports ozone from higher altitudes downward, further increasing surface
ozone concentrations. Compared to the other two typhoon types, Type1 exhibits
weaker ozone formation conditions and better dispersion, resulting in the least severe
ozone pollution.



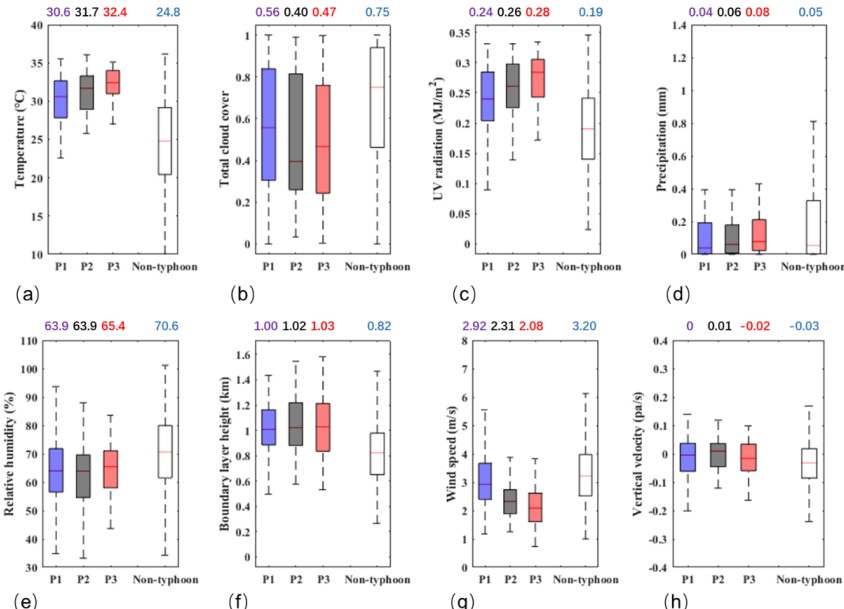


Figure 3. Comparison of meteorological conditions between typhoon and non-typhoon weather.
(a–f) represent 2m temperature, total cloud cover, surface solar radiation, precipitation, relative
humidity, boundary layer height, 10m wind speed, and vertical velocity at 850 hPa, respectively.
P1, P2, and P3 denote three distinct typhoon tracks, while Non-typhoon refers to non-typhoon
conditions. The numerical values above each boxplot indicate the median of the corresponding
dataset.
4.2 Effect of regional transport on ozone distribution
4.2.1 Three-dimensional spatial distribution of ozone
The impact of typhoons on ozone extends beyond creating favorable
photochemical conditions. The regional transport induced by large-scale circulation
plays a pivotal role in determining ozone concentration distribution(Chen et al., 2022b;
Wang et al., 2018). Typhoon tracks modify regional airflow patterns, facilitating cross-
regional transport of ozone and its precursors(Chen et al., 2021). This study employs
three-dimensional reanalysis O$_3$ data (2013-2021) coupled with wind fields and
geopotential height to examine how typhoon-induced regional transport affects the
three-dimensional spatial distribution of ozone concentrations (**Fig4-5**). When the
typhoon moves northward (type 2 and type3), a high-pressure center emerges over





western China at the 175 hPa level, causing a southward displacement of the westerly
jet. Under this circulation pattern, an ozone transport pathway is established,
extending from high to low latitudes and accompanied by subsidence (**Fig. 4b, c**).
Through this transport channel, stratospheric ozone with high concentrations (>75
ppbv) is advected southward to approximately 20°N and descends below the 500 hPa
level (**Fig. 5e, f**). Mechanistic analysis demonstrates that the combined effects of the
westerly jet and mid-latitude high-pressure systems on typhoon motion create upper-
tropospheric wind convergence, which enhances stratosphere-to-boundary-layer
transport of ozone-rich air from the North China Plain (Meng et al., 2022). In contrast,
westward-propagating typhoons (Type 1) do not generate perturbations in the
westerly jet, and no pronounced southward transport or subsidence of upper-level
ozone is evident (**Fig. 5a, d**).

As demonstrated by recent studies (Wang et al., 2022b; Yufeng et al., 2024), the

peripheral circulation of western North Pacific typhoons can effectively transport
ozone and its precursors from source regions (including the Yangtze River Delta, Fujian,
and Anhui provinces) to Guangdong through well-organized atmospheric transport
pathways. Analysis of ozone distribution at the 900 hPa level reveals that northward-
moving typhoons not only induce ozone pollution in Guangdong, but also lead to
elevated ozone concentrations in the Beijing-Tianjin-Hebei and Yangtze River Delta
regions (**Fig. 5b,c**). During the typhoon's northward progression, the low-pressure
center traverses China's eastern coastal areas, where cyclonic circulation facilitates
southward transport of pollutants along the coast, ultimately impacting Guangdong
Province (**Fig. 4e,f**).



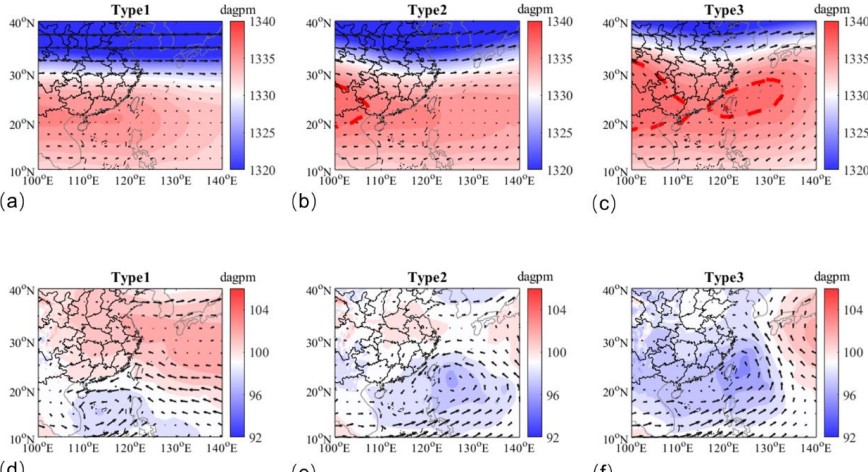


Figure 4. Comparison of circulation patterns under different typhoon tracks. (a-c) show
geopotential height and wind fields at 175 hPa (upper panels) and 900 hPa (lower panels),
respectively. The red curves indicate the positions of high-pressure centers.

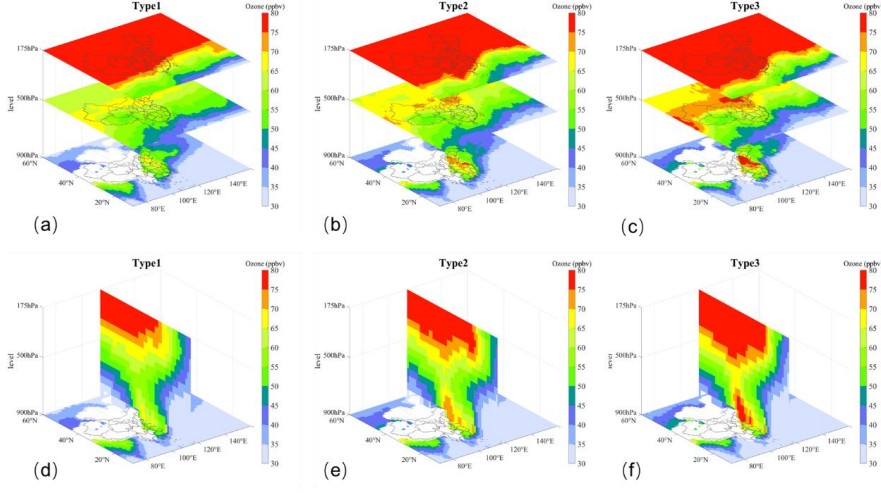


Figure 5. Three-dimensional spatial distribution of ozone under different typhoon tracks. (a-c)
Horizontal ozone distributions at 900 hPa, 500 hPa, and 175 hPa for the three typhoon track
types. (d-f) Horizontal ozone distributions at 900 hPa and corresponding vertical cross-sections
along 114°E for each typhoon type.

**Figure.S5** presents the spatial distribution of ground-level MDA8 $O_3$
concentrations across China, as derived from the reanalysis 1 km high-resolution daily
dataset, under three distinct typhoon track types (type 1, type 2, and type 3). The



analysis focuses on typhoon events, characterized as the date with the highest number
of ground monitoring sites exceeding the 160 μg/m³ (~75ppbv) MDA8 O₃ threshold
during the entire typhoon track.
Being consistent with spatial distribution of ground monitoring O3 concentrations
in section 4.1.2, here reveals significant spatial heterogeneity in O₃ concentrations
across typhoon track types, particularly in Guangdong Province, where the mean
MDA8 O₃ follows the order: Type 2 (56.9 ppbv) > Type 3 (54.6 ppbv) > Type 1 (51.25
ppbv). This variability is attributed to differences in regional transport pathways and
precursor availability. Specifically, type 2 typhoons exhibit elevated O₃ levels in eastern
China but reduced concentrations in northern and central regions compared to type 3.
The enhanced O₃ under type 2 conditions is driven by two synergistic mechanisms: (1)
intensified low-tropospheric transport along China's eastern coastal region, as
evidenced by atmospheric circulation patterns (**Fig. 4e**), and (2) the advection of O₃-
rich air masses from northern and central China, which supply abundant precursors to
Guangdong, particularly its eastern sector. Type 3 typhoons facilitate a more direct,
meridional transport of O₃ from northern and central China, coupled with pronounced
stratospheric intrusions that enhance upper-tropospheric O₃ contributions (**Fig. 5c and
5f**). While type 2 systems lack the robust northern transport pathway observed in type
3, they compensate via secondary O₃ delivery through coastal advection, which
subsequently propagates inland. This dual transport mechanism culminates in the
highest O₃ concentrations in Guangdong, especially the eastern and coastal part,
during type 2 events.
Collectively, integrating atmospheric dynamics (**Fig.4**), three dimensional
evolution of O₃ (**Fig.5**), and ground-level O₃ distributions (**Fig.S5**), underscores the
critical role of typhoon-track-dependent transport pathways in modulating regional O₃
pollution. These highlight the necessity of considering multi-scale meteorological
processes in air quality forecasting and quantifying their contributions to O₃
concentrations across different vertical levels.





To further investigate typhoon-induced ozone variations, spatial ozone
concentration differences between typhoon conditions and non-typhoon conditions
(June-November) were calculated (**Fig.6**). The June-November period was selected to
eliminate seasonal influences. The results indicate that northward-moving typhoons
(Type 2 and Type 3) can induce substantial ozone increases throughout the vertical
column (200-900 hPa) (**Fig. 6b,c**). At the central point (113.23°E, 23.16°N), ozone
concentration changes ranged between 2.5-11.6 ppbv (Type 2) and 0.3-12.3 ppbv
(Type 3). In contrast, Type 1 did not cause significant high-altitude ozone increases,
with central point ozone concentration changes ranging from -3.2 to 0.99 ppbv. Studies
indicate that when gravity waves break in the upper troposphere and lower
stratosphere on the western side of typhoon centers, intense turbulence occurs,
leading to stratosphere-troposphere exchange (STE) (Huang et al., 2024). Subsequently,
typhoons approaching landfall significantly enhance cross-regional ozone transport
from North China to South China through STE (Wang et al., 2024c). This suggests that
after Types 2 and 3 typhoons move northward, their cyclonic circulations transport
high-concentration ozone from the tropopause to lower latitudes and altitudes
through STE, causing significant changes in ozone vertical distribution and increased
ozone concentrations within the boundary layer.





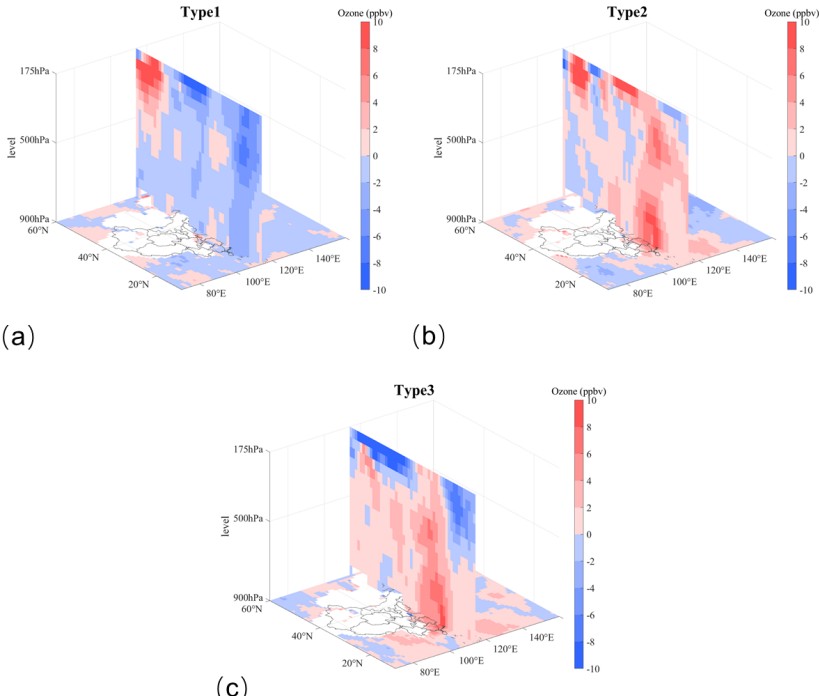

Figure 6. Ozone concentration changes induced by different typhoon types (a-c: horizontal distribution changes at 900 hPa and vertical cross-section changes along 114°E for each typhoon track type respectively).

4.2.2 Boundary layer ozone

To investigate the ozone transport pathways within the boundary layer over Guangdong Province under typhoon conditions, and to examine the differences in ozone sources associated with distinct typhoon tracks, this study conducted HYSPLIT backward trajectory analysis for 237 typhoon events. The analysis focused on 7-day air mass origins at 500m altitude over central Guangdong (**Fig.S6**). For each typhoon type, cluster analysis of air mass origins was performed. After K-value screening, the air mass origins were classified into four trajectory clusters (**Fig.7**). **Table S1** presents statistics for each trajectory type, including: (1) The percentage of different trajectories, (2) mean ozone concentrations along trajectories, and (3) corresponding surface ozone concentrations.



Under Type 1 conditions, air masses in the target area mainly originated from
within the boundary layer, accounting for 60.8%, with air transported from the South
China Sea below 841 meters to Guangdong Province (Traj_4). Analysis of the
subtropical high's influence shows that under this typhoon type, Guangdong
experienced the highest surface pressure and was closest to the subtropical high
(**Fig.S7**). Research indicates that under the influence of the subtropical high, O3
pollution is primarily affected by local emissions (Chen et al., 2024). This aligns with
Traj_4's characteristics of short transport distance and low altitude. The other three
trajectories originated from northwest China (Traj_1, 8.1%), western China (Traj_2,
13.5%), and central China (Traj_3, 17.6%) respectively (**Fig.7a**). The trajectory with the
highest surface ozone concentration was Traj_2, which descended from 3794.1 meters
with an average ozone concentration of 50.3 ppbv along the trajectory(**Fig.7d,**
**Table.S1**). Under Type 2 conditions, nearly half of the air masses in the target area
originated from northwest China (Traj_1, 21.7% and Traj_2, 23.9%), while the other
half came from the South China Sea region (Traj_3, 26.1% and Traj_4, 28.9%)(**Fig.7b**).
Among these, Traj_1 and Traj_2 air masses descended from above 2000m, whereas
Traj_3 and Traj_4 air masses were transported within the boundary layer(Fig.7b). The
trajectory with the highest surface concentration was Traj_1, which descended from
2646 meters with an average ozone concentration of 61.9 ppbv along the trajectory
(**Fig.7e, Table.S1**). Under Type 3 conditions, Traj_1 carried high-concentration ozone
(>75 ppbv) from high-altitude (6356m) over high-latitude areas through North China
to the target region, corresponding to the highest surface ozone concentration (15.2%
proportion) (**Fig.7f, Table.S1**). The other three trajectories originated from central
China (Traj_2, 18.2%) and the South China Sea region (Traj_3, 30.3% and Traj_4,
36.4%)(**Fig.7c**).
A comparative analysis of air mass trajectories from different directions
demonstrates that marine air masses originating from the South China Sea are
characterized by lower altitudes and extended residence time over Guangdong





Province, thereby constituting local ozone pollution sources. Conversely, continental
air masses exhibit longer transport pathways and higher altitudes, representing
regional ozone transport sources. Quantitative analysis reveals that the proportional
contributions of local pollution sources under different typhoon tracks are 60.8%,
55.0%, and 66.4%, respectively. Analysis of long-range transport trajectories reveals
that different typhoon types can respectively deliver ozone from maximum altitudes
of 7,468 meters (~380 hPa), 8,927 meters (~320 hPa), and 9,980 meters (~250 hPa)
into the boundary layer. Type 2 and Type 3 exhibit significantly greater proportion from
upper-level air mass transport (23.9% and 15.2% respectively) compared to Type 1.
These typhoons can transport ozone from altitudes down into the boundary layer.
Combined with the high ozone concentrations along the atmospheric transport
pathways, this results in boundary-layer ozone increases of 10.7 ppbv and 12.3 ppbv
for these two types, respectively (**Fig6.b-c**).


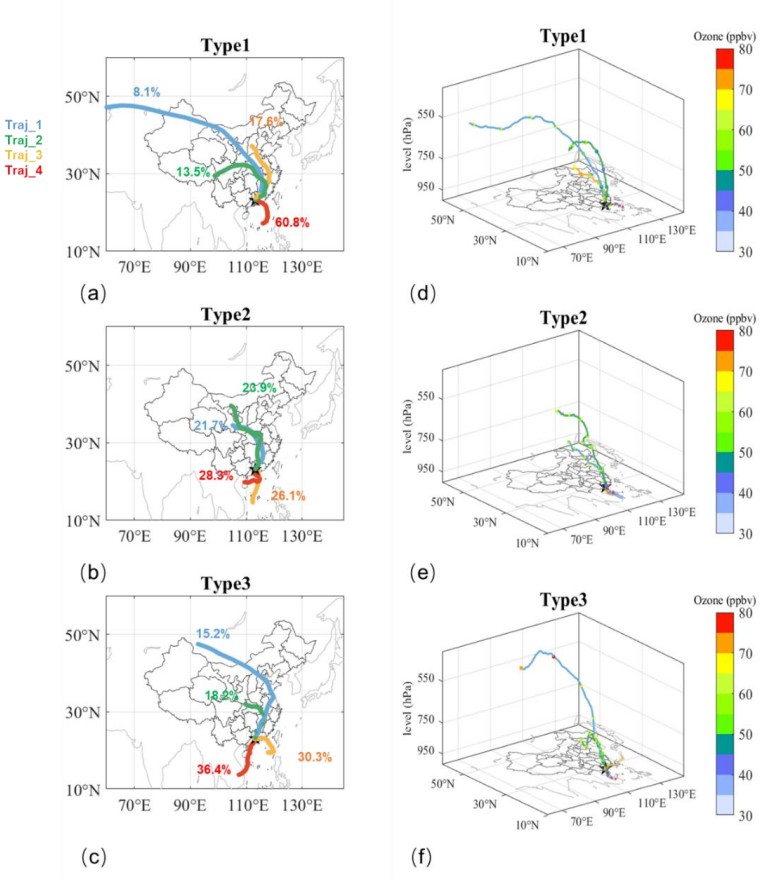


Figure 7. Comparison of boundary-layer air mass trajectory sources under different typhoon tracks: (a-c) two-dimensional views with trajectory percentages indicated numerically; (d-f) three-dimensional views showing ozone concentrations (ppbv) along trajectories (marked by colored points); target regions are denoted by black pentagrams on maps.

4.3 Contribution of typhoons to the vertical transport of ozone
During the period from August 21 to September 6, 2020, the consecutive
occurrence of two northward-moving typhoons (Bavi and Maysak) triggered
prolonged ozone pollution episodes in the Beijing-Tianjin-Hebei and Yangtze River
Delta regions, with over 50% of monitoring stations exceeding ozone standards(Cong
et al., 2024; Hu et al., 2024). Our study reveals that Guangdong Province similarly
experienced extended ozone pollution episodes, particularly between August 28-30



and September 1-3, when more than 40 out of 105 monitoring stations (38.1%)
recorded exceedances. The most severe pollution occurred on August 29, with 57
stations (54.3%) exceeding standards and an average MDA8 ozone concentration of
80.6 ppbv (**Fig.8a-b**). Backward trajectory analysis for August 29 identified a 7-day
vertical transport pathway from upper levels to the boundary layer, suggesting
potential downward mixing of high-ozone air masses (**Fig.8c**). This section examines
the period from August 24 to August 31, 2020, employing the WRF-CMAQ model to
simulate the spatial distribution of ozone. Integrated Process Rate (IPR) analysis is
applied to investigate the formation mechanisms of surface ozone pollution in
Guangdong Province under the influence of consecutive northward-moving typhoons,
with a quantitative assessment of the impact of vertical transport on ozone
concentrations within the planetary boundary layer.
The WRF-CMAQ model was used to simulate ozone variations in Guangdong
Province from August 24 to 31, 2020, with evaluation results showing excellent
performance (**Fig.8d**). For all 105 monitoring stations across the province, the
correlation coefficient between observed and simulated ozone concentrations
reached 0.88 (p<0.01), with a root mean square error (RMSE) of 8.41 ppbv. Focusing
on the Sanshui station (112.8°E, 23.15°N), which exhibited both high ozone levels and
a clear increasing trend, the correlation coefficient was 0.82 (p<0.01) with an RMSE of
3.29 ppbv. These results demonstrate that the WRF-CMAQ model successfully
captured the spatial distribution and temporal evolution of this ozone pollution event
in Guangdong, with statistical metrics meeting operational air quality modeling
standards. The model's strong performance, particularly in reproducing both regional
patterns and local pollution trends, provides reliable support for subsequent analysis
of ozone formation mechanisms under typhoon conditions.





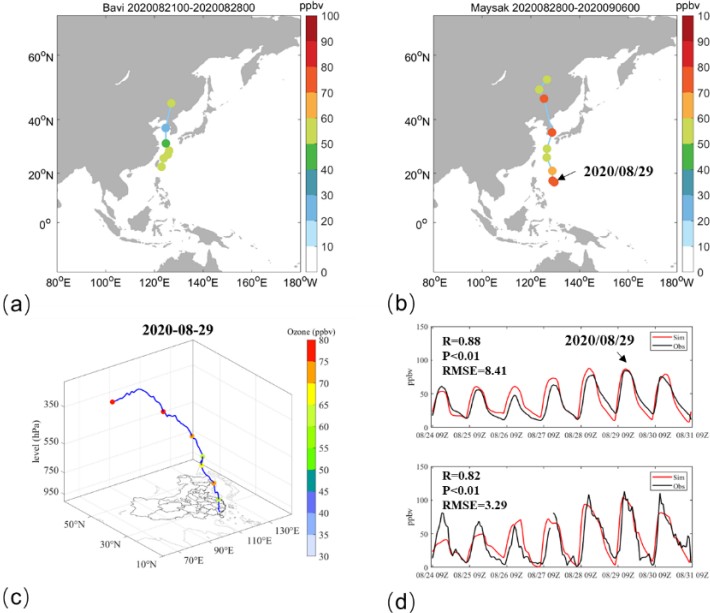


Figure 8. Consecutive northward-moving typhoon tracks, backward trajectories, and ozone
variations. (a-b) Typhoon paths with corresponding MDA8 ozone concentrations in Guangdong
province; (c) Backward trajectories at 1300 LST on August 29, 2020; (d) WRF-CMAQ simulated
ozone variations (upper panel: average across 105 Guangdong monitoring stations; lower panel:
Foshan Sanshui station (112.8°E, 23.15°N) observations, with red lines indicating simulated values
and black lines representing monitored concentrations).

From August 24 to 27, Typhoon Bavi was located along the eastern coastal region
of China, moving northward before gradually dissipating. From August 28 to 31,
Typhoon Maysak emerged in the South China Sea and progressively approached the
Chinese mainland. During this period, we analyzed variations in surface ozone
concentrations and their vertical distribution under the influence of these consecutive
northward-moving typhoons, based on model simulation results (**Fig.9**). The results
show that the variation in surface ozone distribution can be divided into two stages:
The first stage occurred under the influence of Typhoon Bavi, when surface ozone
concentrations rapidly increased in the Beijing-Tianjin-Hebei region, Yangtze River
Delta, and Pearl River Delta, and was rapidly transported to southwestern China by
circulation. The second stage occurred under the influence of Typhoon Maysak, when





ozone concentrations continued to rise across most regions of China. Compared with
the first stage, horizontal ozone transport was not significant during the second stage
(**Fig. 9a-f**). In the vertical dimension, the consecutive northward-moving typhoons
triggered a sustained downward transport process of ozone. Beginning on August 25,
downward ozone transport was observed in the upper atmosphere between 35°N and
40°N. From August 26 to 29, the zone of ozone subsidence gradually expanded
southward, leading to a significant increase in ozone concentrations over Guangdong
Province (**Fig. 9g-l**).

The IPR process analysis results elucidate the impacts of photochemical reactions

and atmospheric transport on ozone concentration variations during this event (**Figs.**
**S8-S9**). The photochemical reactions correspond to the CHEM contribution in the
process analysis. The atmospheric transport represents the combined contributions of
horizontal diffusion (HDIF), horizontal advection (HADV), vertical diffusion (VDIF), and
vertical advection (ZADV) in the process analysis. The results indicate that the increase
in surface ozone was primarily driven by photochemical reactions. During the period
dominated by Typhoon Bavi (August 24-27), photochemical reactions intensified
rapidly over Guangdong Province, contributing more than 30 ppbv to surface ozone
concentrations in the central region (**Figs. S8a-d**). Under the influence of Typhoon
Maysak (August 28-29), the positive contribution from photochemical reactions was
slightly lower than in the previous phase, but still exceeded 16 ppbv in the central
Guangdong region (**Figs. S8e-f**). The contribution of atmospheric transport varied
significantly across different altitudes, exhibiting predominantly negative effects below
850 hPa and positive effects above 850 hPa. Vertical cross-sections of daily mean
atmospheric transport contributions reveal a gradual southward transport of ozone
from higher to lower latitudes. However, its positive contribution to ozone
concentrations was substantially lower than that of photochemical reactions, with
daily mean contributions remaining below 4.5 ppbv (**Fig. S9**). The downward transport
of upper-level ozone inhibited vertical diffusion of surface ozone while simultaneously



transporting high-concentration ozone downward into the boundary layer, further
intensifying ozone pollution levels. In summary, during this ozone pollution event
caused by consecutive northward-moving typhoons: Chemical processes were the
main cause of surface ozone pollution in Guangdong Province, Atmospheric transport
was a secondary contributing factor.

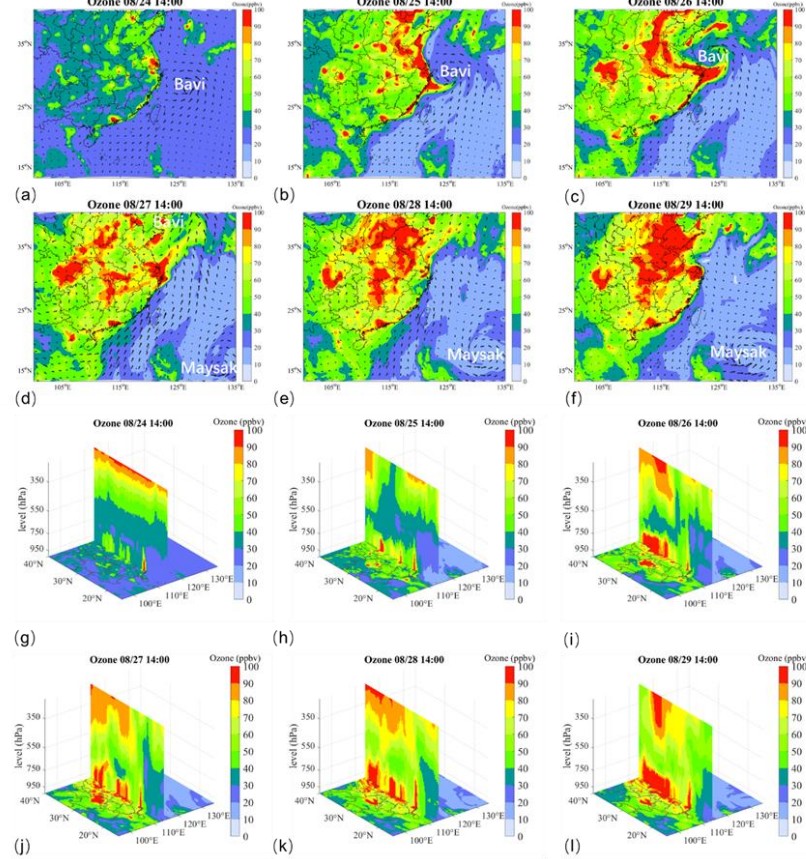


Figure 9. Temporal evolution of (a–f) horizontal distributions of surface ozone and (g–l) vertical
distributions (along 114°E cross-section) of ozone from 1400 LST 24 August to 1400 LST 29 August

2020.

To quantitatively analyze the contribution of vertical transport to ozone
concentrations within the boundary layer, we employed the IPR (Integrated Process
Rate) analysis method to decompose ozone sources and sinks across the study area. A
detailed analysis was conducted using results from the Sanshui station at 100m and





1300m altitudes (**Fig.10a-b**). Subsequently, we calculated the contribution rate of
cross-boundary-layer vertical transport to ozone concentrations in the boundary layer
at each time point (**Fig.10c**) using the following formula (Chen et al., 2022a):
$$Transport\ flux = \left(IPR_{v,pbl} \times Z_{pbl}\ \right) \div \left(\sum_{j=1}^{pbl} O_{3,j} \times Z_j\ \right) * 100\%$$

where $IPR_{v,pbl}$ indicates the IPR value corresponding to vertical transport
(VDIF+ZADV) on the Boundary layer height. that is, the change in the values of
pollutants caused by the vertical diffusion, $Z_{pbl}$ represents the height of the layer in
the model that is close to the height of the boundary layer. $O_{3,j}$ indicates the ozone
concentration in layer $j$ , $Z_j$  represents the height of j layer.
Detailed analysis of process contributions at different heights within the
boundary layer shows that while near-surface atmospheric transport exhibited
negative contributions to daily mean ozone concentrations, the decomposition of
individual processes at 100m height revealed positive contributions from vertical
diffusion (VDIF) during 0900-1100 LST on 29 August, with magnitudes of 39.9 ppbv,
26.4 ppbv, and 12.3 ppbv respectively (**Fig. 10a**). Further analysis of process
contributions at 1300m height reveals distinct positive signals from vertical transport
during the morning hours of both 28 and 29 August (**Fig. 10b**). This confirms that
upper-level ozone can be transported into the boundary layer, thereby influencing
ozone concentrations within the boundary layer. Calculation of cross-boundary-layer
vertical transport contributions revealed six distinct ozone transport events during this
consecutive northward-moving typhoon episode, occurring on 24, 25, 27, 28, 29, and
30 August. The maximum contribution rate to ozone concentrations within the
boundary layer reached 16%.



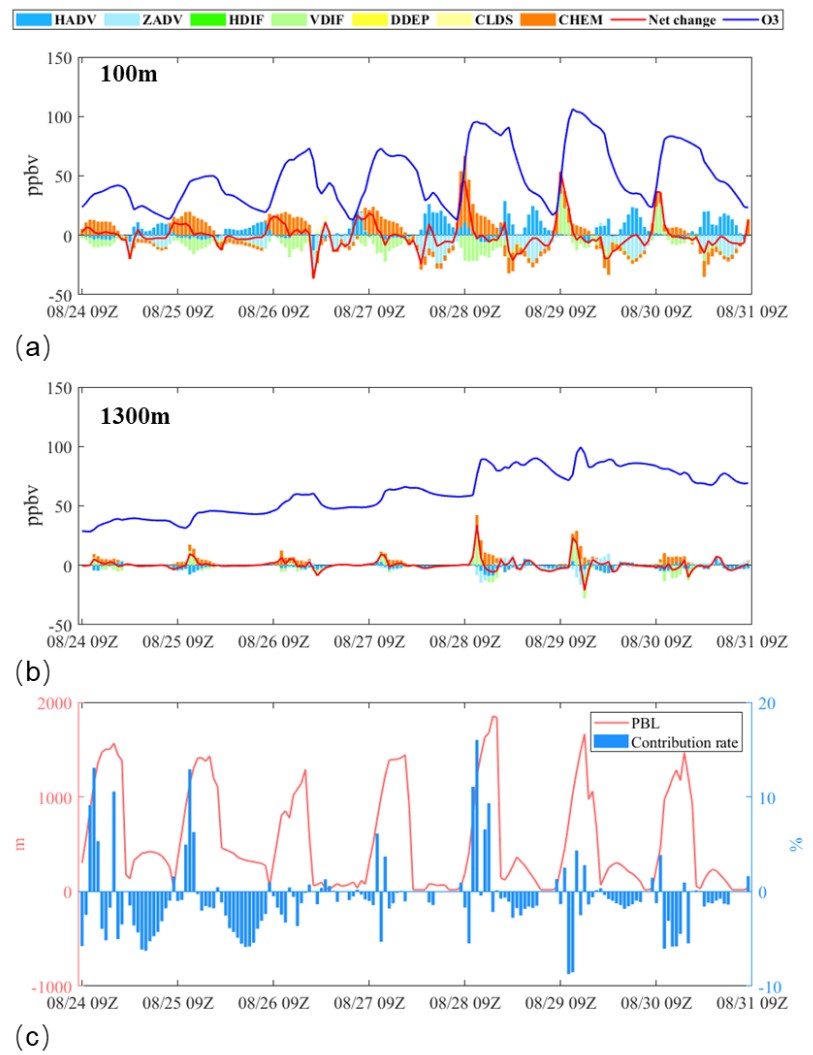

Figure 10. Process contributions to ozone concentrations at 100m and 1300m altitudes, and cross-boundary-layer vertical transport contribution rates. (a-b: Contributions from horizontal diffusion (HDIF), horizontal advection (HADV), vertical diffusion (VDIF), vertical advection (ZADV), chemical processes (CHEM), dry deposition (DDEP), and cloud processes (CLDS). c: Red lines indicate net ozone change, while blue lines show ozone concentration variations. )

**5 Conclusions**

This study systematically investigated the mechanisms by which different typhoon tracks influence ozone pollution in Guangdong Province through meteorological factors, atmospheric circulation patterns, transport trajectories, and



vertical transport contributions, based on 237 typhoons in China's adjacent waters

from 2013-2023. The key findings are:

1.  Historical typhoons were classified into three types using the K-MEANS

clustering method: westward-moving typhoons (Type 1), distant northward-

moving typhoons (Type 2), and proximal northward-recurving typhoons (Type

3). Among these, near-track northward-moving typhoons are more likely to

induce ozone pollution in Guangdong Province due to their more extreme

meteorological conditions, including higher temperatures, stronger solar

radiation, lower cloud cover, reduced precipitation, decreased relative

humidity, elevated boundary layer height, weaker surface winds, and

suppressed vertical motion.

2.  Under the influence of northward-moving typhoons (type2 and type3), an

upper-level anticyclonic center forms near the tropopause height in mid-

latitudes, causing the westerly jet stream to shift southward. This process

triggers the subsidence of high-concentration ozone from the upper

troposphere, accompanied by pole-to-equator transport. Comparative

analysis between typhoon and non-typhoon conditions reveals that both

types of northward-moving typhoons induce significant ozone enhancement

throughout the vertical column, with increases ranging from 2.5 to 11.6 ppbv

(Type 2) and 0.3 to 12.3 ppbv (Type 3).

3.  For Type 1 typhoons, the associated ozone pollution is primarily controlled by

the radiative high-pressure system, with significant contributions from local

pollution sources. In contrast, Type 2 and Type 3 typhoons exhibit the highest

proportions of upper-level transport trajectories (23.9% and 15.2%,

respectively), capable of delivering air masses from as high as 9,980 m (~250

721        hPa) into the boundary layer. Coupled with the elevated ozone concentrations

along these transport pathways, these mechanisms result in ozone

enhancements of 10.7 ppbv and 12.3 ppbv at boundary layer altitudes for



Type 2 and Type 3, respectively.

4. Under the influence of two consecutive northward-moving typhoons from
August 21 to September 6, 2020, Guangdong Province experienced a
prolonged ozone pollution episode. On August 29, ozone exceedance was
observed at 54.3% of monitoring stations. The primary cause of this ozone
pollution event was enhanced photochemical production, with secondary
contributions from upper-level ozone transport. Process analysis revealed
that during 09:00-11:00 LST on August 29, the positive contributions of near-
surface vertical transport to ozone concentrations were 39.9 ppbv, 26.4 ppbv,
and 12.3 ppbv, respectively. During this typhoon event, cross-boundary-layer
transport via vertical mixing contributed up to 16% of the ozone
concentration within the boundary layer.



**Acknowledgments**
The text ends with an acknowledgment section and statement that includes:
• National Natural Science Foundation of China (42121004, 42477273 and
42405194)

• Guangdong Basic and Applied Basic Research Foundation (2023A1515110103 and
2024A1515510025)
• Science and Technology Planning Project of Guangzhou (2025A04J4711)
• Guangdong Province: Special Support Plan for High-Level Talents (2023JC07L057)
• Guangdong Provincial General Colleges and Universities Innovation Team Project
(Natural Science, 2024KCXTD004)

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

Contributions to PM2.5 under Unfavorable Weather Conditions in Guangzhou City,
China, Adv. Atmos. Sci., 35, 1145–1159, https://doi.org/10.1007/s00376-018-7212-9,

884    2018.

Wang, N., Huang, X., Xu, J., Wang, T., Tan, Z. M., and Ding, A.: Typhoon-boosted
biogenic emission aggravates cross-regional ozone pollution in China, Sci. Adv., 8,
https://doi.org/10.1126/sciadv.abl6166, 2022a.
Wang, N., Huang, X., Xu, J., Wang, T., Tan, Z. M., and Ding, A.: Typhoon-boosted
biogenic emission aggravates cross-regional ozone pollution in China, Sci. Adv., 8, 1–
8, https://doi.org/10.1126/sciadv.abl6166, 2022b.
Wang, N., Wang, H., Huang, X., Chen, X., Zou, Y., Deng, T., Li, T., Lyu, X., and Yang, F.:
Extreme weather exacerbates ozone pollution in the Pearl River Delta, China: role of
natural processes, Atmos. Chem. Phys., 24, 1559–1570, https://doi.org/10.5194/acp-
24-1559-2024, 2024c.
Wei, J., Li, Z., Li, K., Dickerson, R. R., Pinker, R. T., Wang, J., Liu, X., Sun, L., Xue, W.,
and Cribb, M.: Full-coverage mapping and spatiotemporal variations of ground-level
ozone (O3) pollution from 2013 to 2020 across China, Remote Sens. Environ., 270,
https://doi.org/10.1016/j.rse.2021.112775, 2022.
Xu, J., Zhou, D., Gao, J., Huang, X., Xue, L., Huo, J., Fu, Q., and Ding, A.: Biogenic
emissions-related ozone enhancement in two major city clusters during a typical
typhoon process, Appl. GEOCHEMISTRY, 152,
https://doi.org/10.1016/j.apgeochem.2023.105634, 2023.
Yang, L., Luo, H., Yuan, Z., Zheng, J., Huang, Z., Li, C., Lin, X., K K Louie, P., Chen, D.,
and Bian, Y.: Quantitative impacts of meteorology and precursor emission changes
on the long-term trend of ambient ozone over the Pearl River Delta, China, and
implications for ozone control strategy, Atmos. Chem. Phys., 19, 12901–12916,
https://doi.org/10.5194/acp-19-12901-2019, 2019.
Yaoyao, C., Tong, L., Yu, W., Jin, S., Yuhong, Z., Siqi, Y., Duohong, C., and Jingyang, C.:
Characteristics of Ozone Pollution in Guangdong Province from 2016 to 2020, 2022.
Ying, M., Zhang, W., Yu, H., Lu, X., Feng, J., Fan, Y., Zhu, Y., and Chen, D.: An Overview
of the China Meteorological Administration Tropical Cyclone Database, J. Atmos.
Ocean. Technol., 31, 287–301, https://doi.org/https://doi.org/10.1175/JTECH-D-12-

913    00119.1, 2014.

Yufeng, Z., Junjun, Y., Tingting, C., Tao, W., Huang, C., Lili, Z., Boguang, W., and
Chengliang, Z.: Influence of typhoon track in northwest Pacific on ozone pollution in
autumn in Shantou City, 2024.
Zhan, C., Xie, M., Huang, C., Liu, J., Wang, T., Xu, M., Ma, C., Yu, J., Jiao, Y., Li, M., Li,
S., Zhuang, B., Zhao, M., and Nie, D.: Ozone affected by a succession of four landfall



typhoons in the Yangtze River Delta, China: major processes and health impacts,
Atmos. Chem. Phys., 20, 13781–13799, https://doi.org/10.5194/acp-20-13781-2020,

921   2020.

Zhu, L., Zhou, R., Di, D., Bai, W., and Liu, Z.: Retrieval of Atmospheric Water Vapor
Content in the Environment from AHI/H8 Using Both Physical and Random Forest
Methods-A Case Study for Typhoon Maria (201808), Remote Sens., 15,
https://doi.org/10.3390/rs15020498, 2023.

