# Peer review of "Investigating the Mechanism of Typhoon Tracks on Ozone Pollution"

_EGUsphere, 2025_

## Author Comment (AC1)

**Summary**

This manuscript systematically investigated the impact of typhoon tracks on ozone pollution in Guangdong, China. It classifies the three type of typhoon paths, quantifies their occurrence frequency and the extent of their impact on ozone, and further elucidates the underlying process mechanisms. Such work is a good supplement to the typhoon–ozone studies. However, some details and explanations need further clarification. I suggest a major revision before the paper can be accepted by Atmospheric Chemistry and Physics. My detailed comments are listed below.

**Major comments**

In the abstract, L53–56 emphasize the role of vertical transport, L56–60 claim that chemical production is the main contributor, and L60–64 again stress the impact of cross-boundary-layer vertical transport. This presentation is somewhat disjointed. The authors should integrate these findings into a logically coherent narrative, rather than simply listing results from the main text, and should clarify which conclusions are drawn from mean-state analyses and which are based on individual cases.

Response:

Thank you very much for your comment. The abstract has been restructured to present photochemical reactions as the primary mechanism and long-range transport as a secondary contributor. This is followed by emphasizing the non-negligible role of long-range transport, with supporting evidence from both case studies and average conditions.The specific modifications are as follows:

**Line 53-67:** The analysis of consecutive northward-moving typhoons' impact on ozone pollution in Guangdong Province reveals that surface photochemical reactions served as the dominant factor, while vertical downward transport of upper-level ozone acted as a secondary contributor. Nevertheless, the long-range vertical transport induced by typhoons cannot be neglected. During this event, vertical transport contributed up to 39.9 ppbv to near-surface (100 m) ozone concentrations, with cross-boundary-layer transport accounting for up to 16% of boundary layer ozone concentrations, demonstrating that typhoon-induced vertical transport significantly enhances boundary layer ozone levels and consequently

worsens surface pollution. Additionally, analysis of the backward trajectories and 3-dimensional ozone concentration fields of 237 typhoons indicates that northward-moving typhoons significantly enhance the downward transport of ozone, consequently increasing the ozone concentration at the central point of Guangdong Province by 2.5 – 14.0 ppbv (Type 2) and 0.3 – 14.5 ppbv (Type 3) within the 250 – 900 hPa layer.

The second and third paragraphs of the Introduction should be more concise, explicit, and logically organized to clearly summarize the progress of previous studies and highlight the specific research gap or problem that this work aims to address.

Response:

Thank you very much for your comment. We have summarized and organized the second and third paragraphs, merging and eliminating some similar and repetitive contents. In addition, the third paragraph has been divided into two parts, highlighting the key scientific issues that need to be addressed in current typhoon research.For details, please refer to **Line 103 - 148**.

L228-229: I am concerned whether setting only 14 vertical layers in the model is sufficient to accurately resolve vertical motions. How many of these layers are within the boundary layer, and how many are in the free troposphere?

**Response:**

Thank you very much for your comment.

The heights of the 14 vertical layers are as follows: 69.39m, 139.21m, 209.50m, 280.35m, 423.23m, 641.40m, 1015.78m, 1485.20m, 1978.85m, 2768.73m, 3735.11m, 5374.64m, 8162.76m, 12181.80m. There are 8 floors below 1500 meters and 6 floors above 1500 meters.

We have determined through the three-dimensional data of ozone and backward trajectory analysis that ozone does indeed transfer downward from the upper layers during typhoon weather. Based on this, we hope to use the model to quantitatively analyze

whether the high-level transportation will affect the interior of the boundary layer. Therefore, more layers were designed near the boundary layer. From the simulation results, it can be seen that this designed layer height can simultaneously reflect the ozone transmission from the upper layer to the lower layer (Figures 9g - l) as well as the ozone variation within the boundary layer (Figure 10). Therefore, we believe that this layer height design is reasonable.

L336–340: It is indeed interesting that a typhoon at such a large distance (even as far north as 60°N) could still influence ozone pollution in the PRD. However, attributing this effect to long-range transport requires supporting evidence.

Response:

Thank you very much for your comment. It is extremely important to us.

The colored dots represent the corresponding ozone concentrations at Guangdong sites along the typhoon tracks. They do not necessarily indicate the high ozone events, represented by red colors, are attributed to typhoons' influence directly.

The red points in Figure 1 at the large distances, such as those in the far north as 60°N, might not indicate persuasive correlation between high ozone in Guangdong and the long-range transport. Instead, it arose from a statistical artifact caused by the temporal overlap of two typhoons, which led to the misattribution of the high ozone values to the latter part of the first typhoon's trajectory. As illustrated in the figure below, Typhoon TRAMI occurred from September 20 to October 3, 2018, while Typhoon KONG-REY spanned from September 28 to October 7, 2018. The former (TRAMI) did not cause ozone pollution in Guangdong Province, whereas the latter (KONG-REY), as it approached the Chinese mainland, triggered severe ozone pollution in Guangdong. However, due to the temporal overlap between September 28 and October 3, the high ozone values were statistically assigned to the later segment of the first typhoon's path. As shown in Figure 9 of the manuscript, the circulation of a typhoon can still induce long-range transport of ozone even when the typhoon is near 35°N. Therefore, we focus primarily on the latitudinal range of 15–40°N. To avoid ambiguity, we have revised the manuscript to clarify the relationship between TYPE2 typhoons and ozone pollution in Guangdong Province.

**Line 337-340**: Typhoons of Type2 can affect the ozone concentration in Guangdong Province from a relatively distant location from the mainland. The paths of typhoons causing ozone pollution mainly fall within the range of 130-150°E and 15-40°N.

[Figure]

[Figure]

Figure. The paths of Typhoons TRAMI (upper) and KONG-REG (bottom) and the corresponding ozone concentrations in Guangdong Province

L430–432: Why would upper-tropospheric convergence lead to stratospheric intrusions into the boundary layer? Does the stratospheric intrusion occur over the North China Plain (NCP)? How does it affect ozone pollution in Guangdong? If L497–501 are intended as an explanation for this issue, they should appear earlier on page 20 to clearly elaborate how stratosphere–troposphere exchange and regional transport contribute to ozone pollution in Guangdong.

Response:

Thank you very much for your comment.

Figure 4c shows a configuration at upper-tropospheric for typhoon track of type 3. This configuration features two distinct, separated anticyclones (the South Asian High (SAH) over the Tibetan Plateau and the western Pacific Subtropical High (WPSH) aloft) with a prominent saddle zone and upper-level convergence between them at 175 hPa. The upper-tropospheric convergence between the highs forces intense large-scale subsidence. This subsidence dynamically pushes the tropopause downward, often leading to tropopause folding on the western flank of the approaching trough associated with the pattern. The fold creates a direct conduit where dry, ozone-rich stratospheric air is injected downward into the upper troposphere.

During the months when typhoons of type 3 track occur, which are summer and autumn months mainly, the stratospheric intrusion usually occurs to the north of 40°N. Stratospheric intrusion is largely attributed to the SAH, WPSH, westly jet stream, and typhoon circulations. It usually occurs over the mid-high latitudes including the NCP. Normally, the stratospheric intrusion in summer and autumn seasons cannot affect ozone pollution in Guangdong significantly. However, stratospheric air from the upper troposphere and lower stratosphere around the periphery of typhoon could guide and enhance the southward and downward transport of stratospheric ozone into the Guangdong province.

The corresponding text in the manuscript is revised as follow:

**LIne 431-449:** Specifically, figure 4c shows a configuration featuring two distinct, separated anticyclones (the South Asian High (SAH) over the Tibetan Plateau and the western Pacific Subtropical High (WPSH) aloft) with a prominent saddle zone and upper-level convergence between them at 175 hPa. Dynamically, such a setup provides a classic

pathway for significant stratospheric ozone intrusion. A tropopause fold, often triggered on the northeastern flank of the SAH where it interacts with the enhanced westerly jet stream, injects a substantial volume of ozone-rich, dry stratospheric air into the upper troposphere. Subsequently, the pronounced subsidence within the saddle zone — induced by the convergence between the SAH and WPSH — effectively transports this intruded air mass downward and southward, directing it toward eastern China and the adjacent oceanic regions. The periphery of typhoon with type 3 tracks could enhance the southward and downward transport of stratospheric ozone, which enhances stratosphere-to-boundary-layer transport of ozone-rich air from the mid-high latitudes (Meng et al., 2022). Through this transport channel, stratospheric ozone with high concentrations (>75 ppbv) is advected southward to approximately 20°N and descends below the 500 hPa level (Fig. 5e, f). In contrast, westward-propagating typhoons (Type 1) do not generate perturbations in the westerly jet, and no pronounced southward transport or subsidence of upper-level ozone is evident (Fig. 5a, d).

L494-495: Does "2.5–11.6 ppbv" and "0.3–12.3 ppbv" refer to column-averaged concentrations? This needs to be clearly stated here as well as in the abstract.

Response:

Thank you very much for your comment. This refers to the ozone variations at an altitude of 250 to 900 hPa. The specific details have been revised in the manuscript. Furthermore, a suggestion from another reviewer prompted an improvement in our seasonal bias removal technique. As a result, the quantified changes in vertical ozone concentration have been modified accordingly.

**Line 504 - 515:** To further investigate typhoon-induced ozone variations, spatial ozone concentration differences between typhoon conditions and non-typhoon conditions were calculated (Fig.6). To eliminate seasonal influences, the anomaly in ozone concentrations between typhoon days and non-typhoon days was first calculated on a monthly basis, after which the different types of statistics were conducted. The results indicate that northward-moving typhoons (Type 2 and Type 3) can significantly increase the ozone concentration at altitudes ranging from 250 to 900hPa (Fig. 6b,c). Within this altitude range,

the variation of ozone concentration at the center point (113.23°E, 23.16°N) changes ranged between 2.5-14.0 ppbv (Type 2) and 0.3-14.5 ppbv (Type 3). In contrast, Type 1 did not cause significant high-altitude ozone increases, with central point ozone concentration changes ranging from -3.5 to 2.5 ppbv.

L666: This expression is not sufficiently rigorous. You seem to be calculating a contribution rate, but the two terms on the right-hand side have different units, and their ratio therefore cannot yield a dimensionless contribution. $IPR_V$, pbl represents a change in concentration, it should be multiplied by the corresponding time interval. Moreover, the left-hand side should not be labeled as a "transport flux," which denotes mass passing through a unit area per unit time.

Response:

Thank you very much for your comment.

Firstly, we have revised "transport flux" to "Contribution rate" (**Line 683**).

Secondly, what we need to explain is that $IPR_{v,pbl}$ represents the change in ozone concentration caused by vertical transport within a unit time (one hour), and its unit is the same as that of ozone concentration. Both sides of this formula consider the concentration change within a unit time, so the time interval is not included.

**Minor comments**

L89: "。" --> ".";

Response: Revised.

L88: "NOx" --> "$NO_x$";

Response: Revised.

L94: You have already defined the abbreviation of $O_3$ at L85.

Response: Revised.

L108: "leads to the formation of elevated ozone concentrations"--> " leads to the elevated ozone concentrations";

Response: Due to the revision of the introduction section, this sentence has been deleted..

L115: "ozone formation efficiency.(Wang et al., 2022a)" --> "ozone formation efficiency (Wang et al., 2022a)";

Response: Revised.

L274: Please check that "degree" or " ° " is used consistently throughout the manuscript.

Response: It has been unified into "°".

L566:How were the ozone concentration values marked along the trajectories in Figure 7 obtained?'

Response: The ozone concentration along the trajectory is derived from TROPESS Chemical Reanalysis $O_3$, and the ozone concentration data at the corresponding positions on the trajectory are extracted.

**Line 537 - 538**: (2) mean ozone concentrations along trajectories (data from TROPESS Chemical Reanalysis O3)

L717: what is the "radiative high-pressure system"?

Response: It has been revised to "subtropical high pressure".

---

## Author Comment (AC2)

This study analyzed the effects of different typhoon tracks on the ozone concentrations through chemical formation, dispersive condition, horizontal and vertical transport using ground-based observations, reanalysis data, and atmospheric chemistry simulation. The analyses and interpretation are sound and reasonable. I only have minor comments.

1. I did not find a data availability section. There are many data included in this study. It would be better to list their public availability.

Response:

Thank you very much for your comment.

In the section 3 of the manuscript, all the sources and URLs of the data are listed in detail. Except for the ozone ground observation data which is not made public, all other data are publicly available.

2. Line 338-339: when the typhoon is distant from the target area, how can the local ozone concentrations be attributed to distant typhoon instead of local generation?

Response:

Thank you very much for your comment. It is extremely important to us.

The colored dots represent the corresponding ozone concentrations at Guangdong sites along the typhoon tracks. They do not necessarily indicate the high ozone events, represented by red colors, are attributed to typhoons' influence directly.

The red points in Figure 1 at the large distances, such as those in the far north as 60° N, might not indicate persuasive correlation between high ozone in Guangdong and the long-range transport. Instead, it arose from a statistical artifact caused by the temporal overlap of two typhoons, which led to the misattribution of the high ozone values to the latter part of the first typhoon's trajectory. As illustrated in the figure below, Typhoon TRAMI occurred from September 20 to October 3, 2018, while Typhoon KONG-REY spanned from September 28 to October 7, 2018. The former (TRAMI) did not cause ozone pollution in Guangdong Province, whereas the latter (KONG-REY), as it approached the Chinese mainland, triggered severe ozone pollution in Guangdong. However, due to the temporal overlap

between September 28 and October 3, the high ozone values were statistically assigned to the later segment of the first typhoon's path. As shown in Figure 9 of the manuscript, the circulation of a typhoon can still induce long-range transport of ozone even when the typhoon is near 35°N. Therefore, we focus primarily on the latitudinal range of 15–40°N. To avoid ambiguity, we have revised the manuscript to clarify the relationship between TYPE2 typhoons and ozone pollution in Guangdong Province.

**Line 337-340**: Typhoons of Type2 can affect the ozone concentration in Guangdong Province from a relatively distant location from the mainland. The paths of typhoons causing ozone pollution mainly fall within the range of 130-150°E and 15-40°N.

[Figure]

[Figure]

Figure. The paths of Typhoons TRAMI (upper) and KONG-REG (bottom) and the corresponding ozone concentrations in Guangdong Province

3. Line 368-370: how would the seasonal biases of meteorological data be eliminated for the comparison between typhoon days and non-typhoon days?

Response:

We appreciate the comment and acknowledge that utilizing non-typhoon days from the entire year for meteorological comparison introduces significant seasonal biases. To address this, we conducted the following analyses:

1. The analysis was conducted in the summer and autumn seasons when the frequency of typhoons was higher and the ozone concentration was also higher. The differences in meteorological characteristics between the periods of June to November (JJA & SON), June to August (JJA), and September to November (SON) were compared respectively.

2. Compare the results in 1 with those before the modification, and analyze which meteorological factors have undergone significant changes after the modification.

And the following conclusions are drawn:

1. Regardless of using the period from June to November (JJA & SON), the period from June to August (JJA), or the period from September to November (SON) to compare typhoon and non-typhoon weather, the results obtained do not conflict with the results using the entire year. The meteorological characteristics of typhoon days are all high temperatures, low cloud cover, high radiation, low precipitation, low humidity, high boundary layer height, low wind speed, and stable weather.

2. There are indeed differences in the numerical values of meteorological factors in summer and autumn, but the overall situation is very similar. To avoid making the article overly lengthy, we decided to use the period from June to November to compare the meteorological differences. Corresponding modifications have also been made in the manuscript, as detailed in Line 372-374 and Line 391-397.

3. By comparing the meteorological conditions before the modification (Before) and after the modification (AFTER), we found that the factors that changed significantly are: temperature (24.8 -> 29.8), radiation (0.19 -> 0.21), precipitation (0.05 -> 0.12), and relative humidity (70.62 -> 71.93).

[Figure]

Figure The changes in meteorological factor values before and after the modification

Table Characteristics of meteorological factors based on different statistical methods

| | Vars | Type1 | Type2 | Type3 | Non-typhoon |
|---|---|---|---|---|---|
| Before: Annual | Temperature (℃) | 30.60 | 31.69 | 32.41 | 24.80 |
| | Total cloud cover | 0.56 | 0.40 | 0.47 | 0.75 |
| | UV radiation (MJ/m^2) | 0.24 | 0.26 | 0.28 | 0.19 |
| | Precipitation (mm) | 0.04 | 0.06 | 0.08 | 0.05 |
| | Relative humidity (%) | 63.97 | 63.88 | 65.44 | 70.62 |
| | Boundary layer height (km) | 1.01 | 1.02 | 1.03 | 0.82 |
| | Wind speed (m/s) | 2.92 | 2.32 | 2.09 | 3.22 |
| | Vertical velocity (pa/s) | 0.00 | 0.01 | -0.02 | -0.03 |
| After: JJA&SON | Temperature (℃) | 30.97 | 32.12 | 32.41 | 29.80 |
| | Total cloud cover | 0.56 | 0.39 | 0.47 | 0.75 |
| | UV radiation (MJ/m^2) | 0.24 | 0.26 | 0.29 | 0.21 |

|  | Precipitation (mm) | 0.04 | 0.06 | 0.08 | 0.12 |
|---|---|---|---|---|---|
|  | Relative humidity (%) | 64.20 | 63.88 | 65.33 | 71.93 |
|  | Boundary layer height (km) | 1.03 | 1.02 | 1.03 | 0.85 |
|  | Wind speed (m/s) | 2.92 | 2.32 | 2.09 | 3.21 |
|  | Vertical velocity (pa/s) | 0.00 | 0.01 | -0.02 | -0.04 |
| JJA | Temperature (°C) | 32.95 | 33.00 | 33.19 | 31.45 |
|  | Total cloud cover | 0.63 | 0.78 | 0.55 | 0.92 |
|  | UV radiation (MJ/m^2) | 0.29 | 0.29 | 0.30 | 0.26 |
|  | Precipitation (mm) | 0.11 | 0.17 | 0.09 | 0.40 |
|  | Relative humidity (%) | 65.63 | 65.26 | 66.23 | 75.08 |
|  | Boundary layer height (km) | 0.96 | 0.93 | 1.03 | 0.86 |
|  | Wind speed (m/s) | 2.66 | 2.22 | 1.98 | 3.60 |
|  | Vertical velocity (pa/s) | -0.03 | -0.03 | -0.02 | -0.05 |
| SON | Temperature (°C) | 29.30 | 30.41 | 30.68 | 26.87 |
|  | Total cloud cover | 0.41 | 0.27 | 0.30 | 0.56 |
|  | UV radiation (MJ/m^2) | 0.23 | 0.25 | 0.26 | 0.18 |
|  | Precipitation (mm) | 0.01 | 0.01 | 0.02 | 0.03 |
|  | Relative humidity (%) | 62.17 | 59.77 | 59.98 | 68.45 |
|  | Boundary layer height (km) | 1.09 | 1.14 | 1.06 | 0.84 |
|  | Wind speed (m/s) | 3.05 | 2.47 | 2.30 | 2.85 |
|  | Vertical velocity (pa/s) | 0.00 | 0.02 | 0.02 | -0.02 |

4. Line 373-374: Why does 14:00 local time be chosen specifically?

Response:

Thank you very much for your comment.

14:00 local time is the period when photochemical reactions are the most intense in a day. We believe that comparing the meteorological conditions at 14:00 is more representative than using the daily average.

5. Line 490-491: How can the selection of June to November help to eliminate the seasonal biases? There are still large variations from summer to fall.

Response:

Thank you very much for your comment.

To eliminate seasonal influences, the anomaly in ozone concentrations between typhoon days and non-typhoon days was first calculated on a monthly basis, after which the different types of statistics were conducted. The results after reanalysis are qualitatively

similar to before, confirming that Types 2 and 3 cause the most notable ozone increases. The specific values have been adjusted, and the revisions are reflected in the manuscript.

**Line 504-515:** To further investigate typhoon-induced ozone variations, spatial ozone concentration differences between typhoon conditions and non-typhoon conditions were calculated (Fig.6). To eliminate seasonal influences, the anomaly in ozone concentrations between typhoon days and non-typhoon days was first calculated on a monthly basis, after which the different types of statistics were conducted. The results indicate that northward-moving typhoons (Type 2 and Type 3) can significantly increase the ozone concentration at altitudes ranging from 250 to 900hPa (Fig. 6b,c). Within this altitude range, the variation of ozone concentration at the center point (113.23 °E, 23.16°N) changes ranged between 2.5-14.0 ppbv (Type 2) and 0.3-14.5 ppbv (Type 3). In contrast, Type 1 did not cause significant high-altitude ozone increases, with central point ozone concentration changes ranging from -3.5 to 2.5 ppbv.

[Figure]

Figure 6. Ozone concentration changes induced by different typhoon types (a-c: horizontal distribution changes at 900 hPa and vertical cross-section changes along 114°E for each typhoon track type respectively).

6. Line 635-636: By the dominance of photochemical reactions, is it dominated by the promoted reaction rates or the promoted precursor concentrations? Generally, how would regional transport of precursor promote the generation of ozone at the target area?

Response:

Thank you very much for your comment.

The IPR analysis in CMAQ can be used to calculate the influence of different atmospheric processes on the values of pollutants, and to quantify the importance of each process in the evolution of the pollutant value. It can only quantify the importance of photochemical reactions, but it cannot distinguish whether it is the increase in the rate of photochemical reactions or the change in the concentration of the precursors that is involved.

Wang et al (2022) combined on-site and satellite observation results with model simulations to find that as the typhoon approached, the cross-regional transport of ozone precursors increased, and under the active photochemical reaction, the ozone formation efficiency increased by more than twice.

Wang, N., Huang, X., Xu, J., Wang, T., Tan, Z. M., and Ding, A.: Typhoon-boosted biogenic emission aggravates cross-regional ozone pollution in China, Sci. Adv., 8, https://doi.org/10.1126/sciadv.abl6166, 2022.